# The Protective Effect of Nutraceuticals on Hepatic Ischemia-Reperfusion Injury in Wistar Rats

**DOI:** 10.3390/ijms241210264

**Published:** 2023-06-17

**Authors:** Carlos Andrés Pantanali, Vinicius Rocha-Santos, Márcia Saldanha Kubrusly, Inar Alves Castro, Luiz Augusto Carneiro-D’Albuquerque, Flávio Henrique Galvão

**Affiliations:** 1Liver and Gastrointestinal Transplant Division, Department of Gastroenterology, University of São Paulo School of Medicine, São Paulo 05403-900, Brazil; cpantanali@gmail.com (C.A.P.); vinicius.rocha@hc.fm.usp.br (V.R.-S.); msk@alumni.usp.br (M.S.K.); profluizcarneiro@gmail.com (L.A.C.-D.); 2LADAF, Department of Food and Experimental Nutrition, Faculty of Pharmaceutical Sciences, University of São Paulo, São Paulo 01246-000, Brazil; inarcastro@gmail.com

**Keywords:** nutraceuticals, liver, ischemia–reperfusion injury, apoptosis

## Abstract

Nutraceuticals are bioactive compounds present in foods, utilized to ameliorate health, prevent diseases, and support the proper functioning of the human body. They have gained attention due to their ability to hit multiple targets and act as antioxidants, anti-inflammatory agents, and modulators of immune response and cell death. Therefore, nutraceuticals are being studied to prevent and treat liver ischemia–reperfusion injury (IRI). This study evaluated the effect of a nutraceutical solution formed by resveratrol, quercetin, omega-3 fatty acid, selenium, ginger, avocado, leucine, and niacin on liver IRI. IRI was performed with 60 min of ischemia and 4 h of reperfusion in male Wistar rats. Afterward, the animals were euthanized to study hepatocellular injury, cytokines, oxidative stress, gene expression of apoptosis-related genes, TNF-α and caspase-3 proteins, and histology. Our results show that the nutraceutical solution was able to decrease apoptosis and histologic injury. The suggested mechanisms of action are a reduction in gene expression and the caspase-3 protein and a reduction in the TNF-α protein in liver tissue. The nutraceutical solution was unable to decrease transaminases and cytokines. These findings suggest that the nutraceuticals used favored the protection of hepatocytes, and their combination represents a promising therapeutic proposal against liver IRI.

## 1. Introduction

Nutraceuticals are natural bioactive or chemical compounds which, in addition to playing a nutritional role, enhance health, cure illnesses, or have preventive properties [1,2]. They are dietary supplements, and from the nutritional point of view, nutraceuticals are a source of nutrients (lipids, carbohydrates, vitamins, proteins, minerals) and non-nutrients (prebiotics, probiotics, phytochemicals, enzymatic regulators) [1,3]. Nutraceuticals can be extracted from both vegetal and animal foods, concentrated, and administered in a suitable pharmaceutical form, with the aim of improving health, in dosages that exceed those obtainable from normal foods [4,5].

In vitro and in vivo studies have provided evidence that nutraceuticals have antioxidant, anti-inflammatory, antibacterial, antiviral, and antifungal activities, as well as evidence that they act as modulators of immune response, angiogenesis, and cell death [4,6,7,8]. These effects are possible due to the multi-target reach of nutraceuticals: endogenous glutathione, interleukins, cytokines, tumor necrosis factor, transcription factor nuclear factor-κB, growth factors, caspases, hepatocyte intracellular neutral lipids, etc. [9,10,11].

By reaching all these targets, nutraceuticals are able to prevent several diseases, such as diabetes mellitus, obesity, cardiovascular diseases, cancer, eye disorders, neurologic diseases, and liver IRI [12,13]. The latter is caused by a limited blood supply and subsequent blood supply recovery during surgical procedures, including management of liver trauma, hepatic resection, and liver transplantation [14,15,16]. It represents the main underlying cause of primary graft dysfunction or non-function and liver failure post-transplantation, in addition to being an important risk factor for acute and chronic rejection [15,17].

Hepatic IRI remains a major unresolved problem in clinical practice [15]. Nutraceuticals are a rising therapy due to their nutritional and therapeutic benefits, as well as safety profile [12]. Some of them have already been studied and exhibited promising results, such as resveratrol, quercetin, omega-3 fatty acids, selenium, ginger, and avocado [18,19,20,21,22,23,24].

Currently, attention has been focused on the synergistic effects of nutraceutical combinations [25,26]. The “synergism concept” was introduced by Liu et al. [27,28]. In this regard, the combination of polyphenols and vitamins is extremely effective in preventing osteoporosis, cardiovascular diseases, cancer, diabetes mellitus, and neurodegenerative diseases [29].

This study aimed to formulate a nutraceutical solution comprising resveratrol, quercetin, omega-3 fatty acids, selenium, ginger, avocado, leucine, and niacin to target the various signaling pathways of liver IRI and decrease its effects. We studied hepatocellular injury, inflammatory mediators, apoptosis by TUNEL, gene expression of apoptosis-related genes, TNF-α and caspase-3 proteins in liver tissue, and histology. We compared the results between five groups: CONTROL—no intervention; IR—rats submitted to liver IRI; NUTRACEUTICALS + IR (NUT + IR)—rats that received the nutraceutical solution by gavage for 7 days and underwent liver IRI; NUTRACEUTICALS (NUT)—rats that received the nutraceutical solution for 7 days; and SHAM—rats submitted only to hepatic manipulation.

## 2. Results

### 2.1. Hepatocellular Injury

The rats of the IR group exhibited a significant increase in serum levels of aspartate transaminase (AST) and alanine transaminase (ALT) compared to the CONTROL and NUT groups. The NUT + IR group presented a significant increase in serum levels of AST and ALT compared to the CONTROL group (Figure 1).

### 2.2. Inflammatory Mediators

There was no difference in terms of IL-1β, IL-6, and IL-10 among the groups. The serum TNF-α level was significantly increased in the IR group compared with the SHAM group (Figure 2).

### 2.3. Lipid Peroxidation

The CONTROL group exhibited a significantly higher level of MDA in the liver tissue when compared to the IR, NUT + IR, and SHAM groups. The same was also observed in the NUT group compared to the NUT + IR group (Figure 3).

### 2.4. Gene Expression of Apoptosis: BAX, BCL-2, CASPASE 8, and CASPASE 3

The gene expression of BAX and BCL-2 was significantly higher in the NUT + IR group compared to the CONTROL and SHAM groups and similar to the NUT and IR groups. The latter group exhibited a significant increase in the gene expression of BAX compared to the CONTROL, NUT, and SHAM groups. Among the gene expression of CASPASES, there was only one difference with CASPASE 3. The gene expression of CASPASE 3 was significantly lower in the NUT + IR group than in the IR group, which in turn had a significantly higher gene expression compared to the CONTROL and SHAM groups (Figure 4).

### 2.5. Immunohistochemistry: Apoptosis, Cleaved Caspase-3, and TNF-α Proteins in the Liver

#### 2.5.1. Apoptosis—TUNEL Assay

The TUNEL assay was used to determinate the apoptosis of the liver cells. The NUT + IR group exhibited a significant decrease in percentage of apoptosis compared to the IR group. Moreover, the IR group had a significantly higher percentage of apoptosis than the CONTROL, NUT, and SHAM groups (Figure 5).

#### 2.5.2. Cleaved Caspase-3 Protein

The immunohistochemistry analysis showed that the cleaved caspase-3 protein in liver tissue was significantly lower in the NUT + IR group than in the IR group. This means that the nutraceutical solution was able to decrease both gene expression and caspase-3 protein in IR injury. The IR group had a significantly higher cleaved caspase-3 protein level compared to the CONTROL and SHAM groups (Figure 6).

#### 2.5.3. TNF-α Protein

In relation to TNF-α protein in liver tissue, the NUT group presented the highest significant percentage compared to all other groups. The IR group had a significantly higher TNF-α percentage compared to the NUT + IR and SHAM groups (Figure 7).

### 2.6. Liver Histological Injury

According to the liver histological score used, the IR group (score 37) had a significantly higher level of liver injury when compared to the NUT + IR group (score 25). It was also observed that the immunohistochemical analysis of caspase-3 showed a marked presence in the related field to IR injury by hematoxylin–eosin (HE), demonstrating a correlation between histological and immunohistochemical findings (Figure 8).

## 3. Discussion

Preexisting nutritional status affects post-operative metabolism, liver function, inflammation, and liver regenerative capacity [30,31]. Therefore, the patient’s condition plays an important role in predicting postoperative complications [13]. Particularly in hepatic IRI, preexisting nutritional status is a major determinant of hepatocyte injury [32].

Several dietary components significantly benefit health, presenting antioxidant or anti-inflammatory properties [31,33]. Hence, the re-establishment and maintenance of correct nutritional status by these nutraceuticals before, during, and/or after surgery could lead to improvements in complications related to IRI. Thus, they represent a potential approach alone or in combination with other therapies to improve patient outcomes [13].

Our nutraceutical solution was unable to decrease the transaminases and cytokines released by liver IRI. One of the probable reasons for that result is the fact that this study was conducted only in the early phase of IRI, not in the late phase when the peak of transaminases and necrosis occur [34,35]. Moreover, the inflammatory mediators reach their peaks at different moments: TNF-α peaks between 30 min and 2 h after reperfusion; IL-1β after 8 h; IL-6 after 12 h; and IL-10 between 30 min and 3 h after reperfusion [36,37,38,39,40]. Therefore, it is necessary to conduct further work to study the effect of nutraceutical solutions on the inflammatory process of hepatic IRI on a timeline.

During liver IRI, Kupffer cells produce reactive oxygen species (ROS) [41]. ROS play a dual role in IRI: they promote apoptosis and stimulate inflammatory mediators, as well as facilitate cell survival under hypoxic conditions and induce antioxidant defenses [42]. In a healthy liver, in response to IRI, levels of PGC-1α, which is a transcriptional co-activator that controls the expression of metabolic pathways, which allow for cellular adaptation to limited nutrient availability, are stimulated, and this stimulation results in increased antioxidant defenses of the cell [43,44]. Supporting this fact, Fukai et al. demonstrated increased total glutathione and reduced glutathione after IRI [45]. This increased antioxidant capacity in non-lethal oxidative stress is one of the mechanisms of the protective effect of ischemic preconditioning [46] and may be the reason why rats in the IR and NUT + IR groups exhibited lower levels of MDA than those in the CONTROL and NUT groups, respectively.

Following the line of not altering the inflammatory process, the nutraceutical solution was able to decrease apoptosis, which is a non-inflammatory subtype of cell death during hepatic IRI [47]. Apoptosis is a form of cell death that is critical in regulating tissue homeostasis, and it is considered the key mechanism of injury during the early phase of hepatic IRI in both experimental and human grafts [48,49,50]. During liver transplantation, apoptosis is involved in cellular injury in acute rejection, as well as in ductopenia seen in chronic rejection [51]. In addition, apoptosis of the donor’s liver is an important predictive factor for early graft dysfunction, and its high rate is associated with shorter graft survival [52].

Furthermore, the histological evaluation adapted from Quireze et al.’s score [53] showed that the nutraceutical solution was also able to significantly reduce tissue injury caused by hepatic IRI. This finding is in accordance with the literature, which has demonstrated that inhibition of apoptosis can decrease IRI in liver grafts [50,54].

After seeing the effect of the nutraceutical solution on apoptosis, we investigated its possible mechanism of action and found that it causes a decrease in gene expression and the caspase-3 protein in liver tissue. These facts may characterize the nutraceutical solution as a *caspase inhibitor*, which involves a novel target to protect the liver from IRI [55]. In this regard, there are some compounds being studied such as IDN-6556 and F573. The pan-caspase inhibitor IDN-6556 inhibits caspase-3 activation and reduces sinusoidal endothelium cell apoptosis when used as an additive in the University of Wisconsin storage solution during the preservation period of rat livers [56]. Moreover, when it is administered in cold storage and flush solutions during human liver transplantation, it provides local therapeutic protection against IR injury and apoptosis [48]. IDN-6556 also protects both murine and human islets in culture and after transplantation, slows down the aminotransferase activity in HCV patients, and lowers portal pressure in patients with compensated cirrhosis and severe portal hypertension [57,58,59].

The other pan-caspase inhibitor F573, in turn, also mitigates liver IRI by reductions in the cytokine TNF-α, apoptosis, and the ALT level [60]. Another application of this caspase inhibitor was shown with the reduction in apoptosis of human and mouse pancreatic islets in vitro and an improvement in their function when they are transplanted into the portal vein [61].

Besides our nutraceutical solution working as a caspase inhibitor, it was also able to decrease the TNF-α protein level in the liver tissue, which is another mechanism of action that may have contributed to decreasing apoptosis. In this regard, Ben-Ari et al. showed that treatment with an anti-TNF-α monoclonal antibody before ischemia is able to mitigate apoptosis by inhibiting the activity of caspases -9 and -3 [62].

There are many other diseases such as inflammatory disorders (psoriasis, arthritis, sepsis), neurologic diseases (Alzheimer’s, epilepsy), metabolic diseases (obesity, diabetes, nonalcoholic liver fatty disease), and cancer that are strongly associated with abnormal activity of caspases and apoptosis [63,64,65,66].

All these studies broaden our horizons and make us think about other possibilities for the clinical use of our nutraceutical combination, in addition to liver IRI. However, before that, there is a need for further studies to establish all the mechanisms of action and the effects of this nutraceutical combination.

Despite the fact that nutraceuticals often pose a challenging pharmacological profile [67], their pharmacokinetic and pharmacodynamic properties are being extended, allowing interpolating results from animals to humans. In this regard, some nutraceutical combinations have already been studied to treat liver steatosis in patients with nonalcoholic liver fatty diseases, showing promising results [68,69].

In addition, these clinical trials are also showing that nutraceutical combinations are being well tolerated and safe [68,69]. Even so, their real efficacy and safety need to be confirmed in future randomized clinical trials.

## 4. Materials and Methods

### 4.1. Animals

Thirty-seven male Wistar rats (245–345 g) were obtained from the Institute of Medical Sciences of the University of São Paulo and housed in LIM37 from the University of São Paulo Medical School. They were placed at room temperature between 20°C and 22 °C, in a 12 h light/dark cycle. The rats were fed with commercial Nuvilab CR-1 Irradiated feed (Nuvital Nutrientes, Colombo, Paraná, Brazil) and hydrated with filtered water ad libitum. The experimental protocol was approved by the Ethics Committee on the use of animals at our institution (909).

### 4.2. Preparation of the Nutraceutical Solution

For the nutraceutical formulation, carboxymethyl cellulose (CMC) syrup 0.5% (Bio Idêntica Manipulation Pharmacy, São José, Santa Catarina, Brazil) was used as a vehicle [70,71].

In the preparation of the nutraceutical solution, omega-3 powder (Natural Products & Technologies, São Bernardo, São Paulo, Brazil) was weighed and transferred to a porcelain mortar, in which 0.5 mL of sunflower oil was added for its solubilization. Then, each remaining component was weighed separately, in the following order: resveratrol, quercetin, chelated selenium, dry ginger extract, avocado powder, leucine, and nicotinamide (Infinity Pharma, Campinas, São Paulo, Brazil). They were mixed with the solubilized omega-3 (Table 1).

The concentration of each nutraceutical ingredient was calculated for a rat with an average weight of 250 g and in a 100 mL solution, without exceeding the toxic limits of each one (Table 1).

The administration of the nutraceutical solution was made at a dose of 1.0 mL, once a day, by gavage (with a BD-12 stainless steel curved gavage needle of 1.2 mm diameter × 41.2 mm length; Ciencor Scientific Ltd., São Paulo, Brazil) for seven consecutive days before the experiment [72,73,74,75].

### 4.3. Anesthesia and Surgical Procedures

The rats were anesthetized with 5% ketamine hydrochloride (Ketalar^®^ Cristália, São Paulo, Brazil) 100 mg/kg and 2% xylazine hydrochloride (Rompum^®^ Bayer, Leverkusen, Germany), at a dose of 10 mg/kg, intraperitoneally. The animals were submitted to orotracheal intubation with a Jelco 16 catheter (Jelco^®^ Descarpack, São Paulo, Brazil) and ventilated with a tidal volume of 0.08 mL/g of weight, a respiratory rate of 60 cycles/min, and a FiO_2_ of 0.21 (Small Animal Ventilator model 683, Harvard Apparatus, Holliston, MA, USA).

A midline laparotomy was performed, and the pedicles of the left lateral and median hepatic lobes were occluded with a 2.5 mm microvascular clamp, inducing the ischemia of approximately 70% of the total liver volume for 60 min [76,77]. The abdominal incision was closed with a continuous 4.0 nylon suture during this period of ischemia to prevent dehydration and hypothermia in animals.

After 60 min of ischemia, the abdomen of the rats was opened again and the clamp was removed to allow for 4 h of liver reperfusion [76,77]. The incision was closed again, and the animals returned to individual cages.

Following the reperfusion time, rats were anesthetized again, and a new midline laparotomy with median thoracotomy was performed. A blood sample was collected through cardiac puncture. Then, the left ventricle was punctured with a Jelco 16 catheter and connected to a 250 mL 0.9% saline solution; the rats were euthanized, and their organs were carefully washed with saline solution [78].

After the liver had been washed homogeneously, a partial hepatectomy of previously ischemic lobes was performed.

### 4.4. Experimental Design

The rats were allocated into five groups. In the CONTROL group (n = 8), the rats did not undergo any surgical procedure. In the IR Group (n = 8), the animals were submitted to hepatic IR. In the NUT + IR group (n = 8), the rats received a nutraceutical solution for seven consecutive days before hepatic IR was performed. In the NUT group (n = 8), the animals received a nutraceutical solution for seven consecutive days. Additionally, in the SHAM group (n = 5), the rats underwent midline laparotomy, and the liver was manipulated without pedicled clamping.

### 4.5. Serum Biochemical Analysis

Serum aspartate aminotransferase (AST) and alanine aminotransferase (ALT) were used as indicators of liver injury. AST and ALT activities were assayed 4 h after reperfusion by ultraviolet kinetic method (COBAS C111, Roche, Indianapolis, IN, USA) according to the International Federation of Clinical Chemistry. The results are expressed in units per liter (U/L).

The following inflammatory mediators were also evaluated: interleukin 1 beta (IL1-ß), interleukin 6 (IL-6), interleukin 10 (IL-10), and TNF-α. Plasma specimens were prepared for analysis in a 96-well plate utilizing a kit of 13-cytokine Milliplex MAP Human Cytokine/Chemokine Magnetic Bead Panel (Millipore Corp., Billerica, MA, USA) following the manufacturer’s recommendations.

Concentrations of cytokines were determined from a standard curve of the mean fluorescence intensity versus pg/mL.

### 4.6. Oxidative Stress

The MDA concentration was determined by reverse-phase High-Performance Liquid Chromatography (HPLC) according to Hong et al. [79]. Liver tissue homogenate (1/40 PBS *v*/*v*) (0.05 mL) was submitted to alkaline hydrolysis with 12.5 µL of 0.2% butylated hydroxytoluene in ethanol and 6.25 µL of a 10 M sodium hydroxide aqueous solution. This mixture was incubated at 60 °C for 30 min, and 750 µL of 7.2% TCA aqueous solution containing 1% KI was added. The samples were kept on ice for 10 min and centrifuged at 10,000× *g* for 10 min. The supernatant (500 µL) was mixed with 250 µL of 0.6% TBA and heated at 95 °C for 30 min. After cooling, the MDA was extracted from the solution with 750µL of n-butanol and analyzed by HPLC (Agilent Technologies 1200 series; Santa Clara, CA, USA). The TBA–MDA conjugate derivative (50 µL) was injected into a Phenomenex reverse-phase C18 analytical column (250 × 4.6 mm; 5 µm, Phenomenex, Torrance, CA, USA) with an LC8-D8 pre-column (Phenomenex AJ0-1287) and was quantified using fluorometric detection at excitation and emission wavelengths of 515 and of 553 nm, respectively [79].

The analysis was run under isocratic conditions, using a mobile phase of 60% phosphate-buffered saline (PBS) (50 mmol, pH 7.1) + 40% methanol at a flow rate of 1.0 mL/min. A standard curve (15–80 µmol MDA, r = 0.9981) was prepared using 1,1,3,3-tetraethoxypropane. The protein concentration was measured by the BCA method using the Pierce BCA kit (Thermo Fisher Scientific, Waltham, MA, USA) as per the manufacturer’s instructions and a solution of bovine serum albumin as standard for the calibration curve (0.025–2.00 mg ptn/mL, r = 0.9969) [79]. Samples were analyzed using a Synergy HT Spectrophotometer (BioTek, Winooski, VT, USA) with Gen5 software version 3.0 (BioTek). The results are expressed as μg MDA/mg protein.

### 4.7. Gene Expression of Apoptosis

Liver tissue from animals in each group was collected, immediately frozen in liquid nitrogen, and stored at −80 °C until RNA extraction was performed. For the extraction of total RNA, TRIZOL™ reagent (Life Technologies Carlsbad, Carlsbad, CA, USA) was used according to the protocol proposed by the manufacturer.

The RNA concentration was determined by the NanoDrop ND-1000 spectrophotometer. The degree of RNA purity was evaluated by a 260/280 nm ratio, using only those whose ratio was ≥1.8. The integrity profile of extracted RNA was evaluated by electrophoresis to verify the presence of bands corresponding to 18S and 28S ribosomal RNAs. The quantified RNA was stored at −80 °C until use.

The design of oligonucleotides was conducted with the Primer 3 program (http://primer3.ut.ee accessed on 22 September 2022). Analysis of the expression of mRNA levels of BAX, Bcl-2, CASPASE 3, and CASPASE 8 genes was performed on a Rotor-Gene RG-3000 thermocycler (Corbett Research, Sydney, Australia). The commercial kit SuperScript™ III Platinum^®^ SYBR Green One-Step qRT-PCR (Life Technologies Corporation, Carlsbad, CA, USA) was used. The beta-actin gene was used as a normalizer of qRT-PCR reactions. The 2-Delta Delta CT method was used for relative quantification of gene expression (Livak & Schmittgen, 2001).

### 4.8. Immunohistochemistry

#### 4.8.1. Apoptosis

For the in situ detection of apoptosis in a single cell, the final identification deoxynucleotidyl transferase (TdT) test was used (TUNEL; Boehringer Mannheim, Germany) [80,81]. According to the standard established by the Laboratory of Histomorphometry and Lung Genomics at the University of São Paulo Medical School, 3–4 µm thick sections of liver tissue were made and placed on silanized slides (Sigma Chemical Co.; St. Louis, MO, USA) on a suitable support, as previously described by Souza et al. [82].

#### 4.8.2. Cleaved Caspase-3 and TNF-α Proteins

Subserial sections from paraffin blocks were used for immunohistochemistry. The antibodies used were caspase-3 and TNF-α (Table 2). Immunohistochemistry was performed according to the manufacturer’s instructions.

Briefly, after the deparaffinization process and the hydration of the liver tissue sections, the recovery of antigenic sites was performed at high temperature in citrate pH 6 for caspase-3 and TRIS-EDTA pH 9 for TNF-α. Endogenous peroxidase blocking was performed with 10 v (3%) oxygenated water four times for 5 min for caspase-3 and for TNF-α with methanol and oxygenated water, volume by volume, two times for 10 min. In the latter two antibodies, protein blots were made, and then the slides were washed in tap water, followed by distilled water, and left in TBS buffer at pH 7.4.

The antibodies were diluted at concentrations shown in the table below (Table 2). The slides were incubated overnight at 4 °C in a humid chamber. Subsequently, incubation was performed with the secondary antibody (ABC Elite, Vector Laboratories Inc, Newark, CA, USA) specific for each species, and the antibody was produced for 30 min in an incubator at 37 °C. Diaminobenzidine (DAB) (Sigma-Aldrich Chemie, Steinheim, Germany) was used as the chromogen. Then, counterstaining was performed with Harris’ Hematoxylin (Merck, Darmstadt, Germany).

### 4.9. Histology

Samples from the median and left anterolateral liver lobes were collected 4 h after reperfusion and fixed in 10% formalin for standard hematoxylin and eosin (HE) staining. A single-blinded pathologist performed the histologic evaluation.

The histologic injury was evaluated according to the scoring system proposed by Quireze et al. [53], which was adapted based on the presence and intensity of the following alterations: ballooning, steatosis, apoptosis, loss of hepatic trabeculae, and necrosis. Those lesions were graded according to the absence (grade 0) or presence of minimal (grade 1), moderate (grade 2), or severe (grade 3) alterations, as determined by the pathologist.

### 4.10. Data Processing

Data were statistically analyzed using GraphPad Prism software (version 9.5.1). One-way analysis of variance (ANOVA) was used to assess differences between tested groups, followed by Tukey’s multiple comparison tests. The non-parametric results of transaminases and gene expression were analyzed by the Kruskal–Wallis test, followed by Dunn’s test. Categorical data of liver histological injury were analyzed using the Chi-square statistic. The results are presented as means ± standard errors of means (SEM). A *p*-value less than 0.05 was considered statistically significant.

## 5. Conclusions

In summary, we proposed a nutraceutical solution that was able to decrease apoptosis and histologic injury caused by liver IRI. Its suggested mechanisms of action are a reduction in gene expression and the caspase-3 protein, as well as a reduction in TNF-α protein in liver tissue. These findings suggest that the nutraceutical combination used favors the protection of hepatocytes and represents a promising therapeutic proposal against liver IRI.

Besides transplantation, hepatocyte apoptosis also occurs in chronic liver diseases that affect 1.5 billion persons globally. All these patients can be helped by controlling apoptosis.

## Figures and Tables

**Figure 1 ijms-24-10264-f001:**
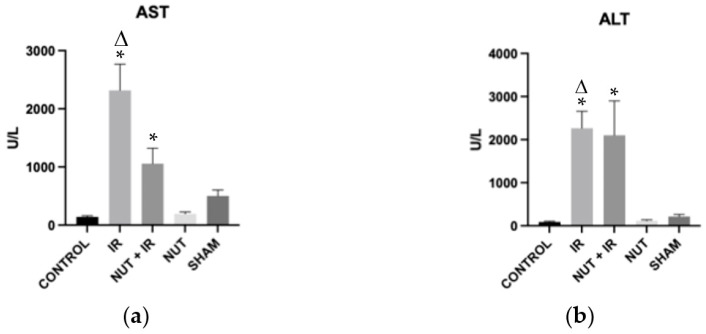
Serum levels of transaminases of each group: (**a**) AST—aspartate aminotransferase; (**b**) ALT—alanine aminotransferase. The data shown are mean ± SEM; * *p* < 0.05 vs. CONTROL group; Δ *p* < 0.05 vs. NUT group.

**Figure 2 ijms-24-10264-f002:**
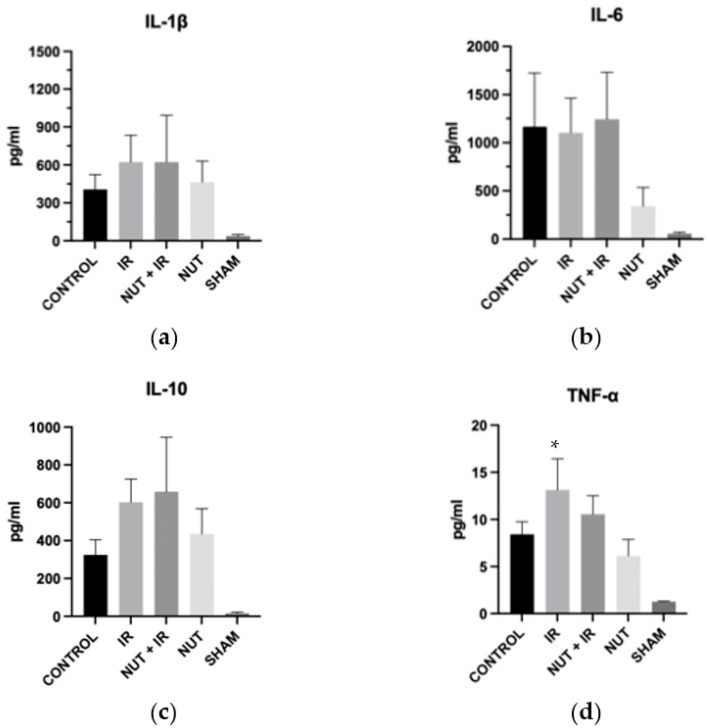
Inflammatory mediators: (**a**) IL-1β, (**b**) IL-6, (**c**) IL-10, and (**d**) TNF-α. The data shown are mean ± SEM; * *p* < 0.05 vs. SHAM group.

**Figure 3 ijms-24-10264-f003:**
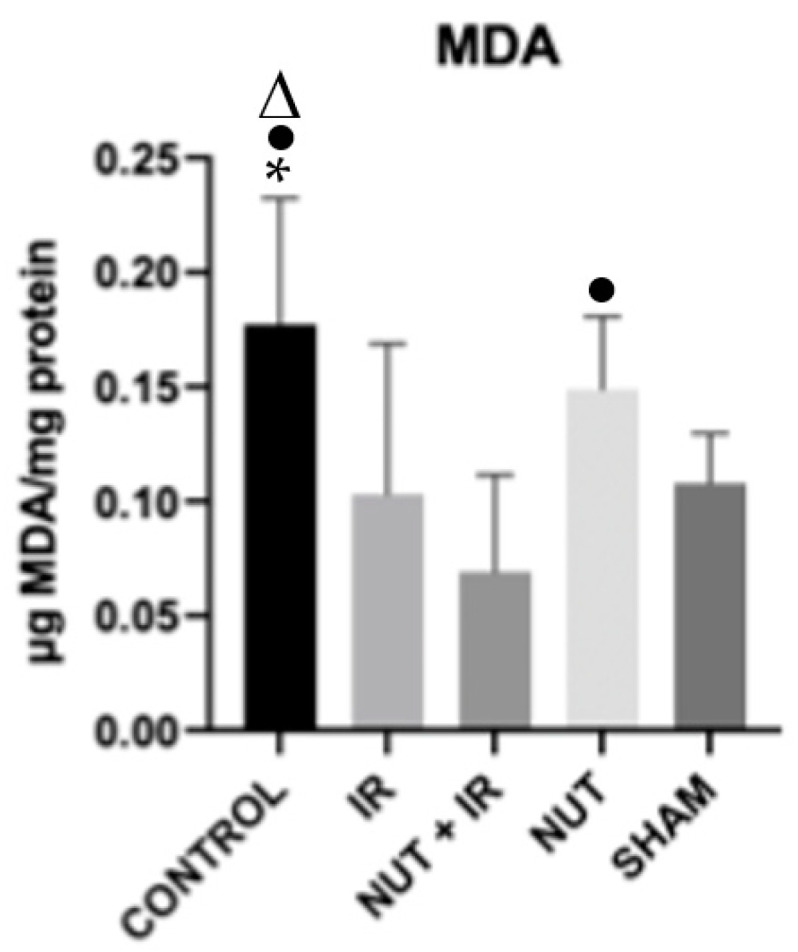
MDA in the liver tissue of each group. The data shown are mean ± SEM; * *p* < 0.05 vs. IR group; ● *p* < 0.05 vs. NUT + IR group; Δ *p* < 0.05 vs. SHAM group.

**Figure 4 ijms-24-10264-f004:**
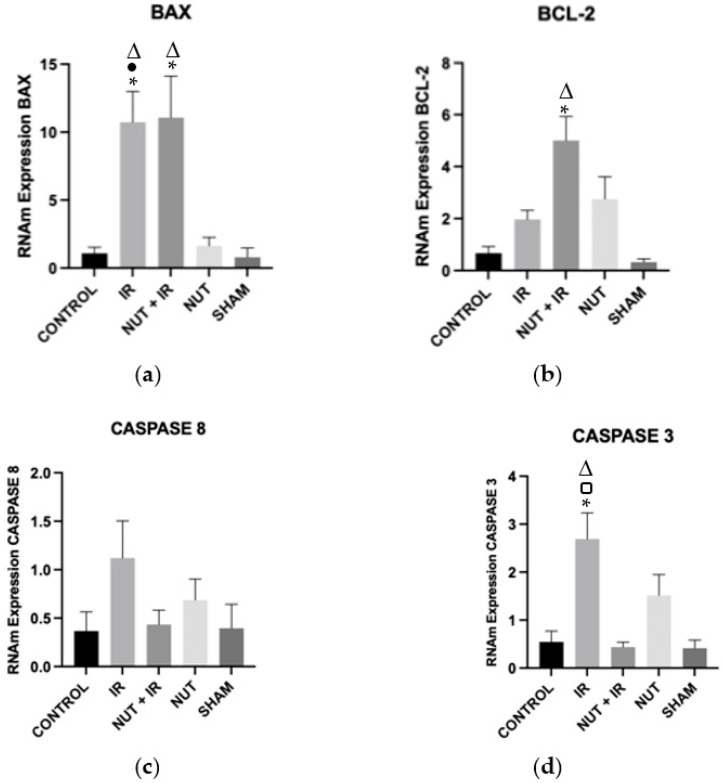
Gene expression of apoptosis-related genes in all groups: (**a**) BAX, (**b**) BCL-2, (**c**) CASPASE 8, and (**d**) CASPASE 3 genes. The data shown are mean ± SEM; * *p* < 0.05 vs. CONTROL group; □ *p* < 0.05 vs. NUT + IR group; ● *p* < 0.05 vs. NUT group; Δ *p* < 0.05 vs. SHAM group.

**Figure 5 ijms-24-10264-f005:**
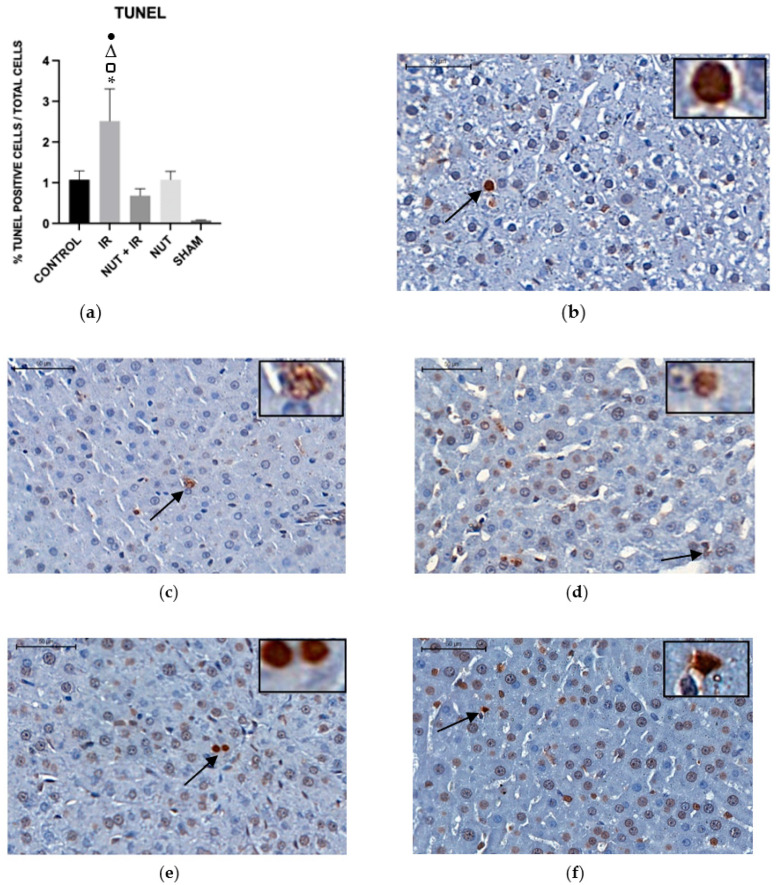
TUNEL assay of hepatic tissues from the different groups: (**a**) data shown are mean ± SEM; * *p* < 0.05 vs. CONTROL group; Δ *p* < 0.05 vs. NUT + IR group; □ *p* < 0.05 vs. NUT group; ● *p* < 0.05 vs. SHAM group. Arrows and highlighted boxes indicate TUNEL positive cells in each group: (**b**) CONTROL; (**c**) IR; (**d**) NUT + IR; (**e**) NUT; and (**f**) SHAM. All images were obtained with 50× magnification and highlighted boxes with 400× magnification, with scale bars of 500 μm and 50 μm, respectively.

**Figure 6 ijms-24-10264-f006:**
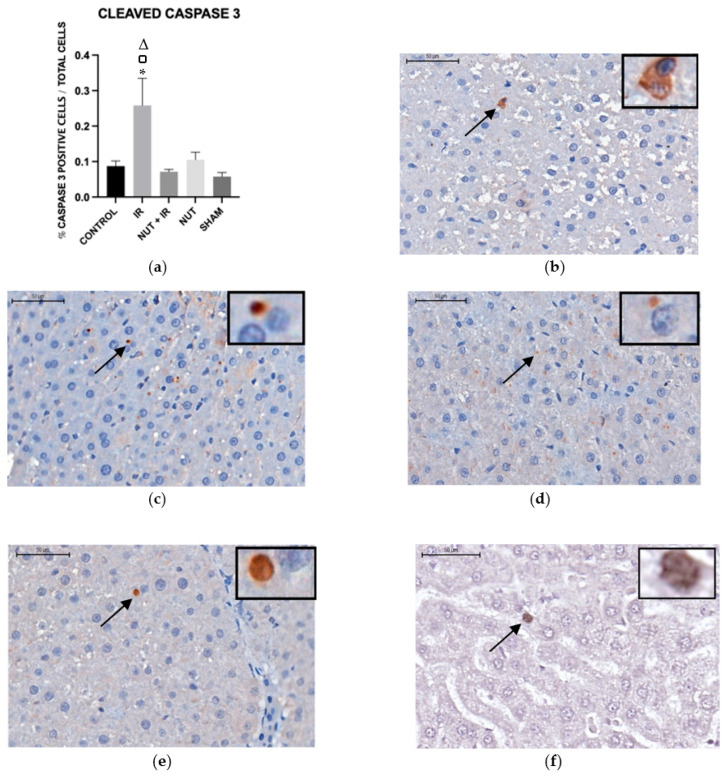
Immunohistochemistry. Cleaved caspase-3 protein in liver tissue from the different groups: (**a**) data shown are mean ± SEM; * *p* < 0.05 vs. CONTROL group; □ *p* < 0.05 vs. NUT + IR group; Δ *p* < 0.05 vs. SHAM group. Arrows and highlighted boxes indicate cleaved caspase-3 positive cells in each group: (**b**) CONTROL; (**c**) IR; (**d**) NUT + IR; (**e**) NUT; and (**f**) SHAM. All images were obtained with 50× magnification and highlighted boxes with 400× magnification, with scale bars of 500 μm and 50 μm, respectively.

**Figure 7 ijms-24-10264-f007:**
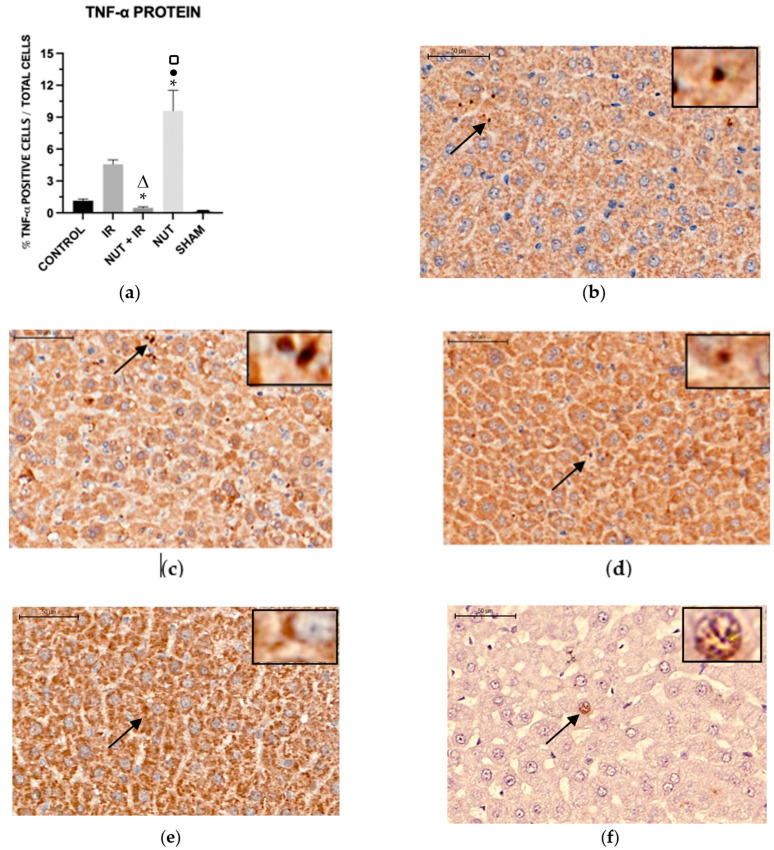
Immunohistochemistry. TNF-α protein in liver tissue from different groups: (**a**) data shown are mean ± SEM; * *p* < 0.05 vs. IR group; Δ *p* < 0.05 vs. NUT group; ● *p* < 0.05 vs. CONTROL group; **□** *p* < 0.05 vs. SHAM group. Arrows and highlighted boxes indicate TNF-α positive cells in each group: (**b**) CONTROL; (**c**) IR; (**d**) NUT + IR; (**e**) NUT; and (**f**) SHAM. All images were obtained with 50× magnification and highlighted boxes with 400× magnification, with scale bars of 500 μm and 50 μm, respectively.

**Figure 8 ijms-24-10264-f008:**
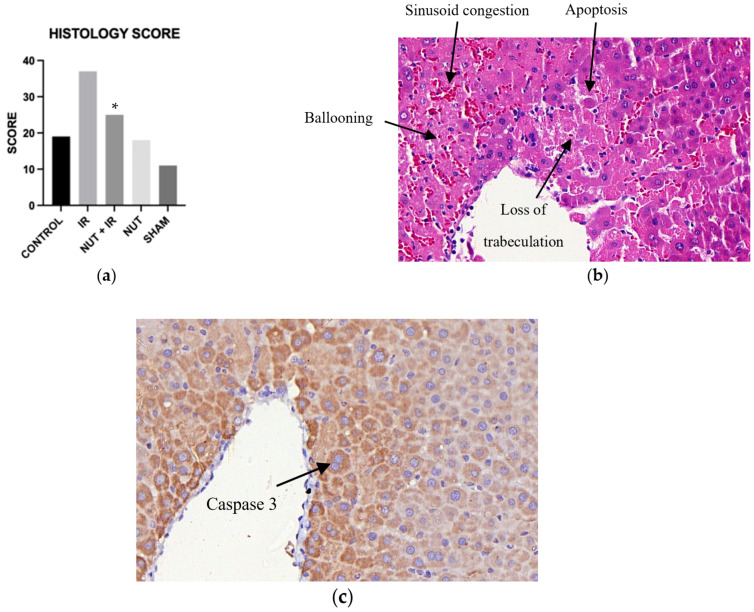
Histological and immunohistochemical analysis: (**a**) total liver histology score from different groups; * *p* < 0.05 vs. IR group. (**b**) HE (Hematoxylin–eosin—600×): photomicrograph of hepatic parenchyma showing ischemic changes (ballooning, apoptosis, destrabeculation, and sinusoid congestion). (**c**) Caspase-3 (Immunohistochemistry—600×): photomicrograph of hepatic parenchyma showing positivity in the cytoplasm (intracytoplasmic brown granular pattern).

**Table 1 ijms-24-10264-t001:** Amount of each nutraceutical.

Nutraceuticals	mg/kg	mg/mL	Amount (g) in 100 mL
Resveratrol	2.96	0.74	0.074
Quercetin	3.56	0.89	0.0908
Chelated selenium	1.76	0.44	2.6831
Omega-3	2.0	0.50	0.05
Ginger extract	3.24	0.81	0.081
Avocado powder	5.08	1.27	0.127
Leucine	4.44	1.11	0.111
Nicotinamide	20.0	5.0	0.5

**Table 2 ijms-24-10264-t002:** Immunohistochemical markers.

Antibody	Concentration	Brand	Code	Clone
Caspase 3	1:200	Novocastra	NCL-CPP32	-
TNF-α	1:200	Santa Cruz	sc-1348	-

## Data Availability

The data presented in this study are available upon request from the corresponding author.

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
