# Peer review of "The Protective Effect of Nutraceuticals on Hepatic Ischemia-Reperfusion Injury in Wistar Rats"

_ijms, 2023, doi:10.3390/ijms241210264_

Round 1

Reviewer 1 Report

In this study, the authors demonstrated that a mix of nutraceutical molecules is able to prevent the liver damage induced by IRI. In this study, the authors focused on the effect of the nutraceutical by evaluating the final effect, i.e. on inflammation and apoptosis, ignoring the possible molecular mechanism.

Please consider referring to these documents and making comments in the introduction and discussion section. PMID: 35986358; PMID: 35185538; PMID: 36014938

Conclusion: Could you please add the public health impact of your findings? What implications?

Reviewer 2 Report

Dear Authors,

I am satisfied with your manuscript entitled “The Power of Nutraceuticals Fighting Liver Ischemia–Reperfusion Injury Through Apoptosis Reduction” This is an interesting article with important new findings of nutraceuticals solution formed by selective nutraceuticals i.e. resveratrol, quercetin, omega-3 fatty acid, selenium, ginger, avocado, leu-16 cine, and niacin. It is well done and the methods and results are clearly described. The discussion and reference are adequate. Some minor comments are as follow

1.       This is an interesting article entitled “The Power of Nutraceuticals Fighting Liver Ischemia–Reperfusion Injury Through Apoptosis Reduction” with important new findings of selective nutraceuticals against Liver IRI. It is well done and the methods and results are clearly described. The discussion and reference are adequate.

2.       What is the criteria for selection of these nutraceuticals only and why chosen these concentration of nutraceuticals in solution?

3.       In the case where the author writes “p<0.05 to p<0.001”in results and tables, it would be good if they could give the actual p value in the text (e.g. p=0.042, p= 0.0009) instead of using this threshold.
